# Patient satisfaction among persons living with HIV/AIDS and receiving antiretroviral therapy in urban Uganda: A factor analysis

**Juliet Nabbuye Sekandi**[1,2]*, **Maria Eugenia Castellanos**[3], **Henok Woldu**[4], **Robert Kakaire**[1], **Simon Mutembo**[5], **Jane Namangolwa Mutanga**[1]

**1** Global Health Institute, College of Public Health, University of Georgia, Athens, Georgia, United States of America, **2** Department of Epidemiology and Biostatistics, College of Public Health, University of Georgia, Athens, Georgia, United States of America, **3** Public Health and Tropical Medicine, College of Public Health, Medical and Veterinary Sciences, James Cook University, Townsville, Queensland, Australia, **4** The Center for Health Analytics for National and Global Equity (C.H.A.N.G.E.), Columbia, Missouri, United States of America, **5** Department of International Health, Bloomberg School of Public Health, Johns Hopkins University, Baltimore, Maryland, United States of America

* jsekandi@uga.edu

**Data Availability Statement:** Data cannot be made publicly available due to ethical restrictions. After closer consultation with the research office at the

## Abstract

### Introduction

Patient satisfaction is an important predictor of health outcomes among patients in HIV/AIDS treatment and care, yet it is rarely measured in routine clinic settings in most of Africa. The aims of our study were to evaluate the internal validity and reliability of the Consumer Assessment of Healthcare Providers and Systems instrument for measuring satisfaction, assess the general level of patient satisfaction, and identify the factors associated with the level of satisfaction among patients receiving antiretroviral therapy in Uganda.

### Materials and methods

We conducted a cross-sectional study of 475 HIV/AIDS-infected patients from July to August 2015 in Kampala, Uganda. Eligible participants were 18 years or older, consented to the study and receiving antiretroviral therapy and outpatient care at the selected public health clinic. This study used a modified version of the validated Consumer Assessment of Healthcare Providers and Systems (CAHPS) instrument to assess the level of satisfaction among HIV/AIDS patients receiving outpatient care. We collected data on socio-demographics, clinical variables and 18-items adapted from the CAHPS instrument rating satisfaction with aspects of health services. We conducted an exploratory factor analysis to assess the internal validity of the 18 items and multiple linear regression analysis of factors associated with patient satisfaction with care.

### Results

Majority of the respondents were females (76.8%), and the mean age was 37 years (SD = 10). The modified CAHPS instrument had high internal consistency (Cronbach's α = 0.94) for measuring satisfaction with HIV/AIDS care. Female sex (p = 0.016), perceived providers'

University of Georgia, the IRB approval for this study restricts the sharing of individual-level data. An anonymized dataset is available upon request from researchers who meet the criteria for access to confidential information. Data requests may be sent to the Human Subjects Office Director at University of Georgia, Kim Fowler (phone contact: 706-542-5318, and email contact: irb@uga.edu). In particular, we welcome researchers willing to create a strong data-sharing partnership and collaboration with the Ugandan researchers who generated the data.

**Funding:** JNS and KO received an internal seed grant from the University of Georgia. The funder provided support in form of funds for clinical research implementation, data collection and salaries for the research assistants. Salary was provided for authors [JNS], but the funder did not have any additional role in the study design, data collection and analysis, decision to publish, or preparation of the manuscript. The specific roles of these authors are articulated in the 'author contributions' section.

**Competing interests:** The authors have declared that no competing interests exist.

technical and interpersonal skills (p = 0.022), emotional health (p = 0.032), and quality of reception services (p<0.001) were significantly associated with satisfaction in this urban HIV/AIDS public clinic.

## Conclusion

The reliability of the CAHPS instrument was high for measuring satisfaction. Providers' technical and interpersonal skills, and the quality of reception services are key to achieving patient satisfaction. Health system interventions to address the gaps identified will enhance the quality of patient-centered HIV/AIDS care in the Ugandan setting.

## Introduction

The UNAIDS estimated that there were 38 million people living with HIV/AIDS infection globally in 2019, with nearly 75% of whom reside in Sub-Saharan Africa [1]. The vision of the global HIV/AIDS response to achieve zero new infections, zero death and zero discrimination cannot be attained without patient-centered care. Uganda has an estimated prevalence of HIV/AIDS of 5.8% among adults aged 15–49 years with a higher of 7.6% for women compared to 4.7% for men in the same age group. About 80% of the 1.46 million people living with HIV/AIDS (PLWH) are receiving care and treatment with antiretroviral treatment [2].

Patient satisfaction can be defined broadly as the 'cognitive and emotional reaction to the components of care delivery and service [3]. Patient satisfaction has been shown to be an indicator of quality of health care services and key predictor of overall health outcomes [4]. In fact, satisfied patients tend to have better adherence to their treatment and continued engagement in care during follow-up visits [5, 6]. Patient satisfaction has been extensively studied in high-income countries, primarily the United States and the United Kingdom [7, 8]. Fewer studies have been done in low-income and there is paucity of evidence especially in HIV/AIDS care [6, 9, 10].

Our previous study done in Kampala, Uganda found that perceived technical competence of the provider and the availability of the services were significant factors associated with general satisfaction in a public HIV/AIDS clinic [11]. Therefore, HIV/AIDS treatment and care programs must address patient satisfaction in order to be successful. As the burden of the HIV/AIDS epidemic continues to affect Sub-Saharan Africa disproportionately, more research is needed to understand the determinants of patient satisfaction in this region [12].

Measurement of patients' satisfaction in routine HIV/AIDS care clinic setting is often lacking especially in Uganda and elsewhere in Africa. There is no universal gold standard instrument for measuring patient satisfaction. A lack of standardized tools that can be used in different specialty care clinic settings is one of the barriers to research in the area of patient satisfaction in Sub-Sahara Africa [4]. The Consumer Assessment of Healthcare Providers and Systems (CAHPS) instrument was developed by the U.S Centers for Medicare and Medicaid Services (CMS) in collaboration with the Agency for Healthcare Research and Quality (AHRQ) in order to provide a standardized survey instrument for measuring patients' experiences with healthcare [13]. The CAHPS Clinician & Group Survey 3.0 (CG-CAHPS) asks patients to report on their experiences with providers and staff in primary and specialty care settings in the last 6 months. The CAHPS tool is unique in that it covers domains of patient experience: accessibility to care, communication with providers, care coordination and interaction with the non-medical staff. It has been widely used in the U.S to assess patient satisfaction across a variety of settings and health conditions [14–16].

Previous studies done on patient satisfaction with HIV/AIDS care in Sub-Saharan Africa have used a variety of measurement instruments [6, 17, 18]. For example, a study done in Zambia adapted the 9-item Adult Primary Care Questionnaire that focuses specifically on the patients' overall satisfaction with their primary care physicians [19]. The limitation of these instruments is that their narrow focus on only the patient-physician or patient-nurse interactions when measuring satisfaction on the day of the visit. In our study, we chose to use the CG-CAHPS instrument because we wanted to capture comprehensive experiences in the last 6 months and with a broader range of clinic staff including the clerks/ receptionists who might indeed influence patient satisfaction yet they are not directly providing medical services. To our knowledge, the CG- CAHPS has not been used in any African population.

The aims of our study were to evaluate the internal validity and reliability of the CG-CAHPS instrument for measuring satisfaction, assess the general level of patient satisfaction, and identify the factors associated with the level of satisfaction among patients receiving antiretroviral therapy at an urban public clinic in Uganda.

## Materials and methods

### Study design, setting and participants

This cross-sectional study was conducted between July 1, 2015 and August 31, 2015. We collected satisfaction data from PLWH attending a public health facility that runs a weekly HIV/AIDS clinic providing free antiretroviral treatment (ART) and care services to over 2,500 people per month. The clinic is one of 14 public health facilities under the Kampala Capital City Authority Health System in the capital of Uganda. The study clinic is specialized for HIV/AIDS care and treatment services. It opens between 8.00 a. m and 5.00 p.m. from Monday to Friday and operates on first come, first served basis therefore patients only receive visit dates but no specific visit appointment timeslots. The clinic has a waiting area which is a semi-closed room equipped with benches seating about 50 patients at a time. The area serves as a reception and registration point where patients present clinic cards with unique identification numbers for retrieval of their paper medical records. In addition, the clinic has 3–4 private rooms that are used for consultation with doctors and health counselors. All registered patients who were 18 years and older, on ART for at least 6 months, able to speak English or Luganda (local dialect), and providing written consent were eligible to participate in the study. Patients were excluded if they were too ill or didn't have enough time to take the 30-minute interview.

### Sample size estimation and recruitment approach

We estimated a final sample size of 450 participants using Kish-Leslie formula with the assumption that at least 50% of patients would report being satisfied with the services provided based on previous published studies on patients' satisfaction in the same setting [11]. We considered a precision of 10%, an alpha of 5% and accounted for a 10% non-response rate. The number of patients to be interviewed were based on the daily average patient load in the study clinic. Based on the records, approximately 113 patients visited the HIV/AIDS clinic each day. We aimed to recruit 20 patients per day in order to achieve the estimated sample size within the planned study enrollment period. We then employed systematic random sampling of every 5th patients who showed up at the clinic on each day of week, Monday through Friday. The clinic nurses assisted the study to team to identify eligible patients and referred them for brief information about the study and then invited to participate voluntarily. If they declined they were replaced by the next patient who met the interval of 5 sampling procedure.

## Data collection instrument and modification of the original CAHPS survey

The psychometric properties of the CAHPS Clinician and Group Survey (CG-CAHPS) instrument have been evaluated in U.S populations and published elsewhere [20, 21]. The original CG-CAHPS 3.0 instrument comprised a total of 24 items [22, 23]. The lead investigator sought input from a team of two local clinicians and three trained research assistants who reviewed the CG-CAHPS version 3.0 questionnaire and agreed on items that were relevant to the HIV/AIDS clinic context in Uganda. Following the team discussions, we modified questions one through 6 to suit the context and then adapted 18-items that were specific to satisfaction. A modified version of CG-CAHPS was created and piloted to assess its suitability for data collection in the HIV/AIDS clinic. Additional questions on socio-demographics such as age, sex, marital status, level of education, rating of overall health and emotional or mental health were added to the final questionnaire. The pilot led to a final version of the adapted instrument which was composed of 35 items which were then translated into Luganda (a local dialect), predominantly spoken by the people residing in Kampala, Uganda (See final questionnaire in S1 Questionnaire). Two bilingual trained research assistants who had vast experience with HIV/AIDS health care but were not working at the study clinic site administered the consent and questionnaire to patients in either Luganda or English.

For this study, the selected 18 items for measuring patient's satisfaction based on relevance to the HIV/AIDS health service context are shown in Table 1. Sixteen of the eighteen items followed a four-point Likert scale (Never, Sometimes, Usually, Always) or (Strongly disagree, disagree, agree, strongly agree), one followed a five-point Likert scale (Excellent, very good, good, fair, poor), and one followed a binary scale (Yes, No). The highest score represented the highest rating for patient satisfaction for a particular health service experience.

For an easy visualization and interpretation of the results, we normalized the scores of each item to a scale of 0 to 100 [22]. Normalizing is a way to transform all scores to the same scale, typically 0 to 100. It is done to ease comparison across items and composites that use different response scales. To transform the scores, we first transformed the response values at the respondent level from 0–100 using the following formula: Normalized Score = 100*(Respondent's selected response value–Minimum response value on scale) / (Maximum response value–Minimum response value). For example, the responses on a four-point scale would be normalized as follows: Response Option 1, 2, 3, 4 and Normalized Response 0.00, 33.33, 66.67, 100.00 respectively. For each participant, we estimated the arithmetic mean of this set of normalized items as a composite index for the independent variable. We estimated measures of central tendency and dispersion for this composite index for the overall population and also stratified by selected variables.

## Exploratory factor analysis methods

As a first step, we computed the polychoric correlation of the 18 original unnormalized items measuring health center experience. Exploratory factor analysis models obtained with this approach have shown to be more consistent with the measured variables than the Pearson correlation when using ordinal data [24]. Then we assessed the appropriateness of our data for factor analysis by performing Kaiser-Meyer-Olkin (KMO) Measure of Sampling Adequacy and Bartlett's test of sphericity [25]. KMO is a test to assess the appropriateness of factor analysis. It evaluates the adequacy of the inter-correlation of a set of variables and each variable for exploratory factor analysis. Typically KMO values less than 0.5 indicate the sampling is not adequate [25].

Of the 18 items initially selected for measurement of satisfaction, three variables were excluded from the analysis because they had low KMO (<0.40) and poor correlation with the

**Table 1. Items adapted /modified from Consumer Assessment of Healthcare Providers and Systems (CAHPS) Clinician and Group Survey.**

| No. Question | Original Question | Display of question in this manuscript | Original Response Option |
|---|---|---|---|
| 6 | In the last 6 months, when you came to this facility during regular office hours, how often did you get an answer to your medical problem that very day? | Patient gets an answer to the medical problem the day of the visit | Never<br>Sometimes<br>Usually<br>Always |
| 7 | Wait time for care includes time spent in the waiting room and exam room. In the last 6 months, how often did you see this provider within one hour of your arrival? | Provider sees patient within one hour of arrival | Never<br>Sometimes<br>Usually<br>Always |
| 8 | In the last 6 months, when you needed to see a provider, how often did you see a provider as soon as you needed? | Provider saw patient as soon as needed | Never<br>Sometimes<br>Usually<br>Always |
| 9 | In the last 6 months, when you came to receive your medical treatment, did you feel comfortable while waiting in the facility? | How comfortable patient feels while waiting in the facility | Never<br>Sometimes<br>Usually<br>Always |
| 10 | In the last 6 months, how often did this provider explain things in a way that was easy to understand? | Provider explain things in an easy way | Never<br>Sometimes<br>Usually<br>Always |
| 11 | In the last 6 months, how often did this provider listen carefully to you? | Provider listen carefully to patient | Never<br>Sometimes<br>Usually<br>Always |
| 12 | In the last 6 months, did you talk with this provider about any health questions or concerns? | Patient talk about provider about their concerns | Yes<br>No |
| 13 | In the last 6 months, how often did this provider give you easy to understand information about these health questions or concerns? | Provider gives patient easy to understand information | Never<br>Sometimes<br>Usually<br>Always |
| 14 | In the last 6 months, how often did this provider seem to know the important information about your medical history? | Provider knows about patient medical history | Never<br>Sometimes<br>Usually<br>Always |
| 15 | In the last 6 months, how often did this provider show respect for what you had to say? | Provider shows respect for what patient say | Never<br>Sometimes<br>Usually<br>Always |
| 16 | In the last 6 months, how often did this provider spend enough time with you? | Provider spends enough time with patient | Never<br>Sometimes<br>Usually<br>Always |
| 18 | I am willing to recommend this facility to family and friends | Patient willing to recommend this facility to family and friends | Strongly disagree<br>Disagree<br>Agree<br>Strongly agree |
| 19 | I am willing to return to this facility for care next time | Patient willing to return to this facility for care next time | Strongly disagree<br>Disagree<br>Agree<br>Strongly agree |
| 20 | I am willing to adhere to my medical regimen | Patient willing to adhere to my medical regimen | Strongly disagree<br>Disagree<br>Agree<br>Strongly agree |

(*Continued*)

**Table 1.** (Continued)

| No. Question | Original Question | Display of question in this manuscript | Original Response Option |
|---|---|---|---|
| 21 | Overall, would you rate the quality of care and services received during visits at this facility | Quality of care and services received | Excellent<br>Very good<br>Good<br>Fair<br>Poor |
| 22 | In the last 6 months, how often were clerks at this facility as helpful as you thought they should be? | Clerks are helpful | Never<br>Sometimes<br>Usually<br>Always |
| 23 | In the last 6 months, how often did clerks at this facility treat you with courtesy and respect? | Clerks treat patient with courtesy and respect | Never<br>Sometimes<br>Usually<br>Always |
| 24 | In the last 6 months, how often were the clerks at this facility efficient from check-in through check-out? | Clerks are efficient from check-in to check-out | Never<br>Sometimes<br>Usually<br>Always |

other variables, therefore 15 items were included in the final exploratory factor analysis (S1 Table). The overall KMO result of the selected variables was 0.94; high values (between 0.5 and 1.0) indicate appropriateness. The significance of the Bartlett's test (p-value < 0.0001, Chi Sq. = 8461.51 and df = 105) also showed that the correlation matrix was not an identity matrix and satisfies the criteria for factor analysis [26].

We determined the number of factors to retain for the final analysis by examining a scree plot, which is a graphical representation of the eigenvalues of the correlation matrix. We considered factors with eigenvalues of 1 or greater and factors that explain at least 10% of variability with a combined cumulative variability of 60% or greater [27, 28]. The correlation matrix was factor analyzed using maximum likelihood using Promax rotation (package *psych* version 1.9.12.31) [29]. We examined the loading pattern of the rotated factors to determine the factors that have the most influence on each item. After identification of the factors, each of them was labeled and a factor score was created by adding the normalized items that were grouped in a given factor [30].

## Reliability

To assess the internal consistency (reliability) of the patient satisfaction survey instrument, we calculated the Cronbach's alpha coefficient for the 15 items in the modified CAHPS questionnaire and for the factor scores. Conventionally, Cronbach's alpha greater than 0.70 is deemed to reflect satisfactory reliability [31]. In addition to the reliability coefficient for each factor, the mean and standard deviation of each normalized item under each factor was calculated.

## Association of patients and providers factors with satisfaction rating scores of the health facility

To identify patients and providers factors that influence patients' perceptions of satisfaction with health care, we used normalized rating scores as the outcome variable (0 for the lowest and 100 for the highest rating). Independent variables included in the analyses were primarily based on the bivariate analysis and also on literature review. These were sex, age, education level, marital status, and duration of care and the mental and emotional health of the patients.

A multiple linear regression analysis model was used to assess the association of factor scores that affect patient's rating of the health facility. Beta coefficients, standard errors and p-values a significance level of <0.05 are reported. A positive coefficient indicates an increase in rating associated with a unit increase of the corresponding item. We used R-squared ($R^2$) to assess the model's adequacy and $\alpha = 0.05$ to determine the significance level. Data were analyzed using R version 3.6.0.

### Ethics approval and consent to participate

The protocol and informed consent documents were evaluated and approved by the Institutional Review Boards or Ethics Committees of the University of Georgia and the School of Public Health at Makerere University. Once local approval was obtained, the project was reviewed and approved by the Uganda Council for Science and Technology. Administrative permission was granted by the HIV/AIDS public health clinic.

## Results

We enrolled 475 PLWH receiving antiretroviral therapy at a public health clinic in Kampala, Uganda. The non-response rate was 5%, mostly comprising of patients who stated that they did not have time to participate in the survey. The majority of the respondents were females (76.8%) and the mean age was 37 (SD = 9.8) (Table 2). Fifty-three percent of patients had at least a primary level of education and 43.9% were currently married. Almost 90% of the patients had attended the public health clinic for more than one year and 87% of them reported having visited the clinic at least 4 times. The overall mean score of general patient satisfaction was 66 (SD 18) (Table 2). The mean satisfaction scores were higher for men (70, SD 18) compared to women (65, SD17). Mean satisfaction scores did not differ much by age, education, marital status and duration in HIV/AIDS care. The mean scores were higher among patients who reported more clinic visits. As one would expect, there was a consistent increase in mean satisfaction scores with better ratings of general overall health and mental/emotional health. This reflects face and construct validity of the results. The scores were also very similar across the categories which may indicate high collinearity between the variables hence the decision to include only one of them in further analyses.

Responses from over 50% of patients reflected inadequate quality services for at least four items. Most patients (83%) said "never" or "sometimes" when asked if patients got an answer to their medical problem the day of the visit, 71% responded "never" when asked if patients talked to provider about health questions or concerns. Majority (62%) also reported "never or sometimes" when asked if patients saw provider within one hour of arrival at the clinic and 53% of patients "never" or only "sometimes" saw their provider as soon as needed (Fig 1).

### Exploratory factor analysis and development of the factor scores

The scree plot of the 15 items included in the exploratory factor analysis suggested a four-factor solution (Fig 2). We selected a three-factor solution as it explained 77% of the variability because the addition of a fourth factor to the solution increased the explained variance by only 4%.

Factor 1, corresponding to Technical & Interpersonal Skills factor accounted for 35% of the variability in patient satisfaction, while Willingness to Adhere and Quality of Reception Service accounted for 22% and 20% respectively. Factor 2 showed high positive loadings on the item related to specific providers' attributes such as technical competence, communication and interpersonal skills labeled as "Technical & Interpersonal Skills" (Table 3). Factor 2 had high positive loadings on the item related to the clerk's helpfulness, courtesy and efficient in check-

**Table 2. Baseline characteristics and general satisfaction scores (scale 0–100) of 475 people living with HIV/AIDS on antiretroviral therapy at a public clinic in urban Uganda.**

| Characteristics | Frequency (%) | Normalized mean score (SD) |
|---|---|---|
| General Satisfaction | 475 | 66 (18) |
| Sex | | |
| Male | 110 (23.2) | 70 (18) |
| Female | 365 (76.8) | 65 (17) |
| Age (Years) | | |
| 18–29 | 118 (24.8) | 64 (18) |
| 30–39 | 172 (36.2) | 66 (17) |
| 40–49 | 125 (26.3) | 65 (18) |
| > = 50 | 60 (12.6) | 69 (18) |
| Education | | |
| No Education | 59 (12.4) | 68 (17) |
| Primary 1–7 | 253 (53.3) | 65 (17) |
| Senior 1–6 | 148 (31.2) | 66 (17) |
| College and above | 15 (3.2) | 67 (27) |
| Marital Status | | |
| Single | 28 (5.9) | 64 (19) |
| Currently Married | 208 (43.9) | 67 (18) |
| Previously Married | 238 (50.2) | 65 (17) |
| Duration of Care | | |
| <1 year | 55 (11.6) | 60 (14) |
| 1–3 years | 219 (46.1) | 69 (17) |
| >3–5 years | 90 (18.9) | 61 (17) |
| > 5 years | 111 (23.4) | 65 (18) |
| Number of visit times | | |
| 2 to 3 times | 62 (13.1) | 54 (12) |
| 4 times | 214 (45.1) | 63 (17) |
| 5 to 9 times | 137 (28.8) | 73 (17) |
| 10 or more | 62 (13.1) | 70 (17) |
| Self-rate Overall Health | | |
| Poor | 5 (1.1%) | 48 (9) |
| Fair | 122 (25.7%) | 53 (13) |
| Good | 215 (45.3%) | 62 (15) |
| Very good | 114 (24.0%) | 77 (14) |
| Excellent | 19 (4.0%) | 81 (15) |
| Self-rate Overall Mental or Emotional Health | | |
| Poor | 12 (2.5%) | 47 (7) |
| Fair | 160 (33.7%) | 55 (14) |
| Good | 198 (41.7%) | 65 (15) |
| Very good | 97 (20.4%) | 76 (15) |
| Excellent | 8 (1.7%) | 81 (16) |

in/check- out services and this labeled as "Quality of Reception Service". Factor 3 had high positive loadings on the item related to willingness of the patients to adhere to their treatment, return to the clinic or refer others to use the same service, labeled as "Willingness to Adhere" (Table 4).

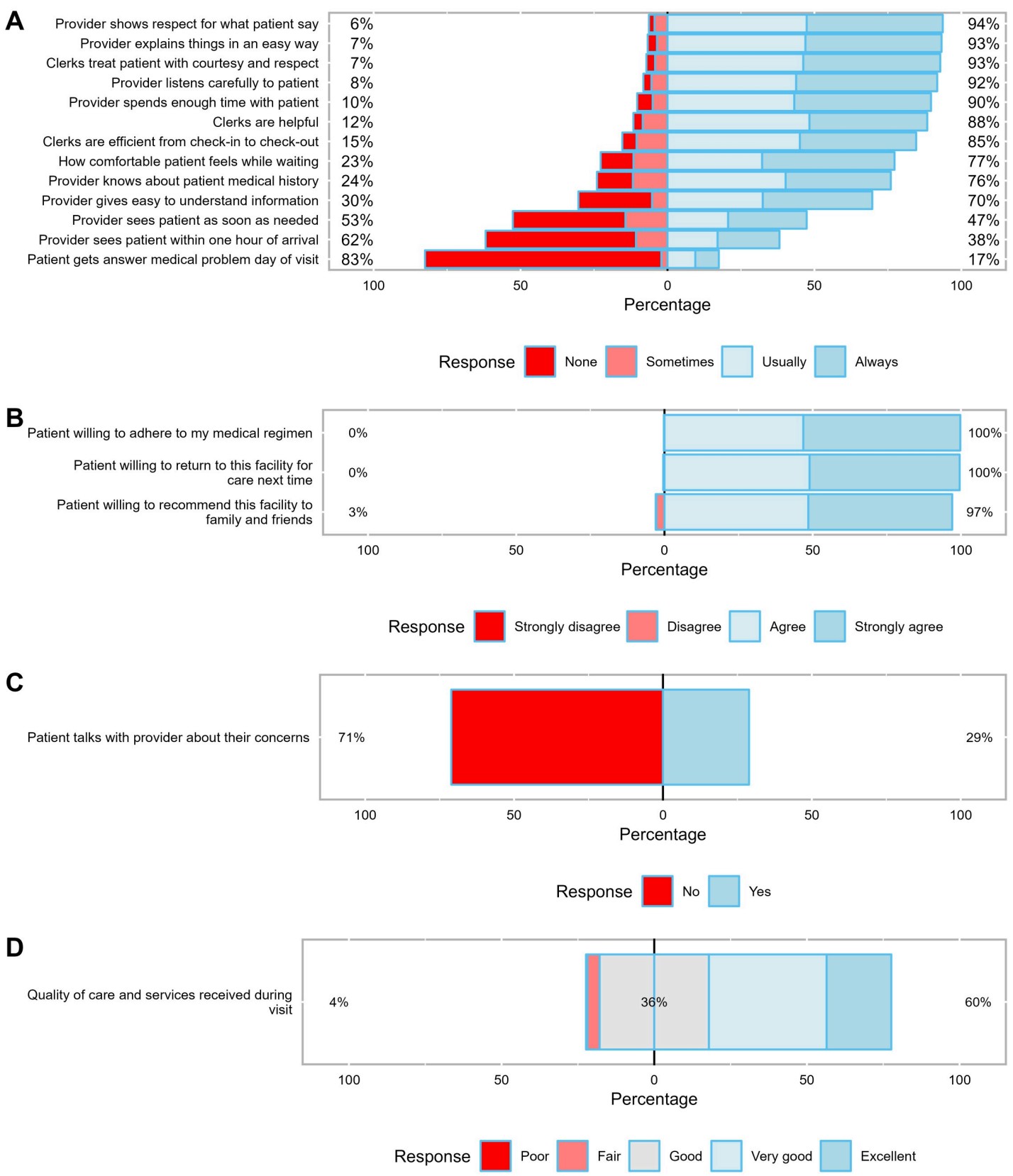

**Fig 1. Responses from 6-month survey measuring Patient Satisfaction with Health Services for 18-items from 475 people living with HIV/AIDS on Antiretroviral Therapy at a Public Clinic in Urban Uganda (Four panels: A-D).**

### Internal consistency

Overall, internal consistency was very high with Cronbach's alpha coefficient of 0.94. Additionally, the internal consistency remained high across the three factors solutions at 0.92 for the Technical & Interpersonal Skills factor, 0.90 for the Willingness to Adhere factor and 0.89 for the Quality of Reception Service factor (Table 5). The question with the lowest mean value was how often the provider saw the patient within one hour of arrival (mean score = 36, SD = 41) and the items with the highest mean values were the ones related to willingness of the patient to adhere.

### Bivariate and multivariable regression analyses

Technical & interpersonal skills and quality of service factors were significantly associated with patient satisfaction after controlling for age, sex, education, marital status, mental and emotional health of the patient and, duration of care (Table 6). This suggests that providers' technical and interpersonal skills as well as patients' perceptions of service quality are strong predictors of satisfaction with HIV/AIDS service in this public health clinic setting. On average, patient's satisfaction rating was higher among females than males (estimated β coeff. = 0.412, p-value = 0.016) and on patients with higher mental/emotional health (estimated β coeff. = 0.206, p-value = 0.022). We also observed that having attained senior education level was independently associated with lower satisfaction ratings of the HIV/AIDS clinic (β coeff. = -0.515, p-value = 0.034) but it was not statistically significant in the adjusted regression analysis.

### Discussion

In this study, we evaluated the internal validity and reliability of the CAHPS instrument, the level of general satisfaction and the associated factors among people living with HIV/AIDS

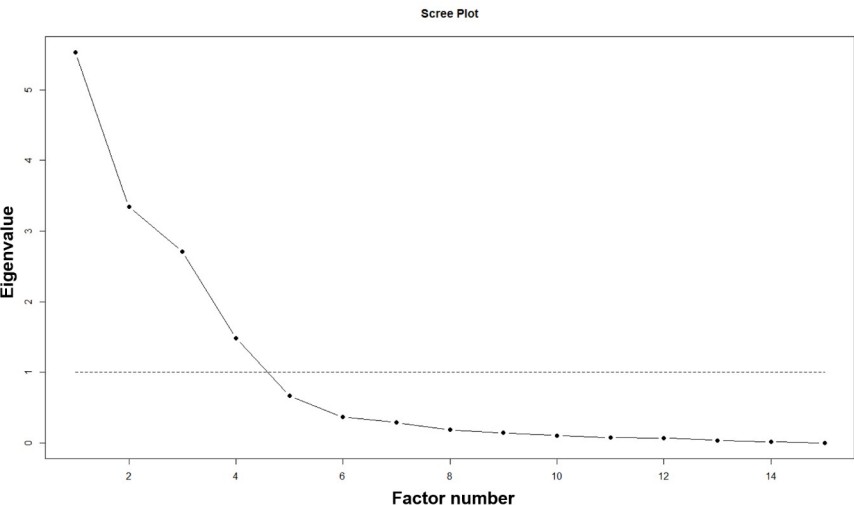

**Fig 2. Scree plot showing the number factors for 15-items and corresponding eigenvalues for 475 patients at public clinic in urban Uganda.**

**Table 3. Factor analysis of survey items on Patient Satisfaction with Health Service in 475 people living with HIV/AIDS on antiretroviral therapy at a public clinic in urban Uganda.**

| Items Used in Factor Analysis | Factor 1 [Interpersonal & Technical Skills] | Factor 2 [Willingness to adhere] | Factor 3 [Quality of service] |
|---|---|---|---|
| How often did the provider give you easy to understand information about health questions or concerns | **0.86** | -0.18 | 0.18 |
| How often did the provider know about your medical history | **0.82** | -0.01 | 0.09 |
| How often did the provider show respect for what say | **0.76** | 0.24 | -0.06 |
| How often did the provider listen carefully to you | **0.76** | 0.31 | -0.16 |
| How often did the provider explain things in an easy way | **0.75** | 0.28 | -0.11 |
| How often did you see a provider as soon as you needed | **0.61** | -0.11 | 0.35 |
| Feel comfortable while waiting in the facility | **0.58** | -0.05 | 0.15 |
| How often did the provider spend enough time with you | **0.54** | 0.39 | 0.05 |
| How often did you see a provider within one hour of arrival | **0.41** | 0.10 | 0.34 |
| I am willing to adhere to my medical regimen | -0.07 | **0.97** | 0.10 |
| I am willing to return to this facility for care next time | 0.00 | **0.93** | 0.01 |
| I am willing to recommend this facility to family and friends | 0.09 | **0.65** | 0.19 |
| How often were clerks efficient from check-in to check-out | -0.09 | 0.22 | **0.82** |
| How often were the clerks as helpful as you thought | 0.04 | 0.07 | **0.82** |
| How often did the clerks treat you with courtesy and respect | 0.23 | -0.04 | **0.75** |

Bold face loadings under each column represent the items (questions) that were factored to create that specific factor represented by the conventional name assigned to it.

receiving care in Uganda. We found that the instrument was highly reliable to measure patient satisfaction as indicated by Cronbach's alpha of 0.94. The mean score for level of general satisfaction was modest and the factors associated with the patient's rating of the clinic were provider interpersonal & technical skills and quality of healthcare services. To our knowledge, this is the first study to assess the reliability of the CAHPS instrument in this African population. This study builds of the very few studies that have systematically assessed patient satisfaction among PLWH on ART in Uganda and in Africa [6, 17, 32–35].

Patients were satisfied with items related to the respect and treatment provided by the providers and clerks. As other studies have reported, patients were not satisfied by the speed of accessing the service nor did they felt that they can discuss health questions or concerns with the provider [36]. However, patients overwhelmingly wanted to return to the clinic and would recommend this facility to friends and family signaling that despite the perceived shortcomings, patients valued the service provided by this urban clinic.

When we examined the three domains identified by the factor analysis, we found that quality of service had the strongest association with the rating of the health care facility. Two variables had the highest loadings (efficiency of the clerks for the check-in and the checkout and how helpful the clerks were at the visit), indicating the important contribution of these variables to this domain [37]. Factor loadings specifically show how well the items cluster together.

**Table 4. Total variability in patient satisfaction explained by three factors.**

| Factor | % of Variability Explained | Cumulative % of Variability |
|---|---|---|
| Factor 1-Interpersonal & Technical skills | 35% | 35% |
| Factor 2- Willingness to Adhere | 22% | 57% |
| Factor 3- Quality of Reception Service | 20% | 77% |

**Table 5. Cronbach's alpha reliability of three factors and normalized mean scores (scale 0–100) of patient satisfaction in HIV/AIDS care.**

| Patient Satisfaction Item | Normalized mean score (SD) |
|---|---|
| **Overall (Cronbach's alpha = 0.94)** | |
| **Factor 1-Technical skills (Cronbach's alpha = 0.92)** | |
| How often did the provider give you easy to understand information about health questions or concerns | 61 (39) |
| How often did the provider know about your medical history | 67 (33) |
| How often did the provider show respect for what say | 79 (22) |
| How often did the provider listen carefully to you | 79 (24) |
| How often did the provider explain things in an easy way | 79 (23) |
| How often did you see a provider as soon as you needed | 45 (41) |
| Feel comfortable while waiting in the facility | 71 (33) |
| How often did the provider spend enough time with you | 77 (26) |
| How often did you see a provider within one hour of arrival | 36 (41) |
| **Factor 2-Willingness to adhere (Cronbach's alpha = 0.90)** | |
| I am willing to adhere to my medical regimen | 84 (17) |
| I am willing to return to this facility for care next time | 83 (17) |
| I am willing to recommend this facility to family and friends | 82 (19) |
| **Factor 3- Quality of Reception Service (Cronbach's alpha = 0.89)** | |
| How often were clerks efficient from check-in to check-out | 75 (25) |
| How often were the clerks as helpful as you thought | 79 (23) |
| How often did the clerks treat you with courtesy and respect | 73 (27) |

**Table 6. Unadjusted and adjusted linear regression coefficients of mean satisfaction and patients' characteristics.**

| | Unadjusted | | | Adjusted | | |
|---|---|---|---|---|---|---|
| Item | β Estimate | SE[1] | p-value | β Estimate | SE[1] | p-value |
| Age (years) | 0.005 | 0.007 | 0.528 | 0.002 | 0.007 | 0.810 |
| Sex-Female | 0.288 | 0.171 | 0.093 | 0.412 | 0.170 | **0.016** |
| Education | | | | | | |
| Primary level | -0.252 | 0.227 | 0.269 | -0.136 | 0.209 | 0.516 |
| Senior level | -0.515 | 0.242 | **0.034** | -0.292 | 0.226 | 0.197 |
| Tertiary level | -0.739 | 0.454 | 0.105 | -0.491 | 0.438 | 0.262 |
| Marital Status | | | | | | |
| Currently Married | 0.025 | 0.318 | 0.936 | 0.048 | 0.299 | 0.872 |
| Previously Married | 0.230 | 0.315 | 0.467 | 0.166 | 0.303 | 0.584 |
| Duration of Care | | | | | | |
| 1–3 years | -0.127 | 0.236 | 0.590 | -0.342 | 0.227 | 0.133 |
| >3–5 years | -0.442 | 0.268 | 0.099 | -0.342 | 0.250 | 0.172 |
| > 5 years | 0.233 | 0.258 | 0.367 | 0.151 | 0.245 | 0.537 |
| Mental/Emotional Health Patient | 0.386 | 0.086 | **<0.001** | 0.206 | 0.089 | **0.022** |
| Factor1-Interpersonal &Technical Skills | 0.002 | 0.000 | **<0.001** | 0.001 | 0.000 | **0.032** |
| Factor2- Willing to Adhere | 0.006 | 0.001 | **<0.001** | -0.003 | 0.002 | 0.096 |
| Factor3-Quality of Reception Services | 0.009 | 0.001 | **<0.001** | 0.007 | 0.001 | **<0.001** |

Footnote

[1]SE = Standard Error. Response is Rating of the health care facility (Ranging from 0- lowest to 10-highest rating) A positive β coefficient indicates an increase in rating of health care associated with a unit increase in corresponding continuous variable or a change in level for categorical covariate.

Our findings highlight the importance of the timeliness with reception services and courteous treatment of patients by the clerical staff in the HIV/AIDS clinics. Previous studies have shown that if the clerical staff are friendly, they can improve the patient trust and satisfaction with the facility [38].

Technical and interpersonal skills of the providers was another domain associated with a positive rating of the health care facility. This domain comprised nine items, with the highest loadings from the item 'how often did the provider give the patient easy to understand information about health questions or concerns?' and 'how often did the provider know about the patient medical history?' These findings suggest that patients appreciate when the providers give them more personalized attention, remembering who they are and providing them with information that is easy to digest in relation to their concerns [32, 39]. A study done in Tanzania reported similar findings when evaluating satisfaction with services by doctors and nurses at two HIV/AIDS care clinics [35]. Another study done in Zambia among patients who were lost-to-follow up showed that patients who were satisfied with their HIV/AIDS providers were significantly more likely to re-engage in care than those who were not satisfied [6]. These findings highlight the importance of continuously improving provider–patient relationships in HIV/AIDS care and treatment programs.

Other patient characteristics that were identified as significantly associated with rating of the clinic included being female and patients with high mental/emotional health. Similar findings for sex differences in satisfaction have been described in Vietnam and Cameroon [10, 17]. The patients' perceived health status has been shown to affect the patient's satisfaction level with quality of care. Studies show that patients with poor self-reported health tend to feel less satisfied with their healthcare [40, 41]. Having attained up to the senior level of education was negatively associated with an increase in the satisfaction ratings of the clinic but this association was not statistically significant at multivariable regression analysis. A previous study we conducted in Mulago outpatients' clinics in Kampala, Uganda showed that patients with higher education were less satisfied than those with a lower education level [11]. A possible explanation could be that more educated patients are more knowledgeable and thus have higher expectations of the quality of the service they receive. In contrast, a study done in Zambia found that HIV/AIDS patients with no formal education most commonly expressed non-satisfaction [6]. This contrast in findings could arise from differences in the methods used to measure satisfaction or the populations studied.

Our study results should be interpreted in light of some limitations. First, we surveyed a convenience sample of HIV/AIDS outpatients receiving care at one urban public clinic. The findings may not be generalizable to all urban clinics or private HIV/AIDS clinics or those in the rural settings. However, the results can generally inform the design of future research and interventions to address service delivery gaps. Secondly, we used a cross-sectional design which represent a snapshot in time. We acknowledge that patients' satisfaction can vary over time depending multiple factors. Therefore, we cannot directly infer causality between the factors we examined and satisfaction. In future, prospective cohort studies should be conducted to capture the time varying nature of patients' satisfaction across visits. Social desirability bias could have possibly occurred given that participants were being asked to rate the same facility where they received services. However, we minimized this bias by hiring a separate research team of nurses who do not routinely work at the same HIV/AIDS clinic. Additionally, patients were assured that their information was to be kept confidential and only shared in a general report as recommendations for improvement of quality of services. Finally, recall bias could have affected the measure of satisfaction which was based on the self-reported information about the last 6 months. Selection and sampling bias could also affect our results since the majority of participants were female. We attempted to minimized the bias by using systematic sampling.

The practical implications of the study findings point to the need for continuous quality assessment and improvement interventions at a local and regional level to ensure that patients are satisfied with the care they receive at public health facilities from check-in to check-out. Enhancement of relevant technical, interpersonal and client care skills can be achieved by periodic in-service training and performance appraisals that are tailored to needs in professional development.

## Conclusions

This study showed that the adapted CAHPS instrument was reliable for measuring patient satisfaction among people living with HIV/AIDS receiving ART. The factor analysis highlighted provider technical and interpersonal skills as well as the patients' perceived quality of reception service as significant predictors of satisfaction in patients receiving HIV/AIDS services. Health system interventions to address these gaps should be explored to improve health care quality in HIV/AIDS outpatient clinics in this setting.

## Supporting information

**S1 Table. Responses from 6-month survey measuring Patient Satisfaction with Health Services for 18-items from 475 HIV/AIDS-infected patients on antiretroviral therapy at a public clinic in urban Uganda.**
(DOCX)

**S2 Table. Individual and overall Kaiser's measure of sampling adequacy.** Initial results with 18 items and final selection with 15 items included in the exploratory factor analysis.
(DOCX)

**S1 Questionnaire. CG-CAHPS 6-months survey questions in English final version.**
(PDF)

## Acknowledgments

The authors acknowledge Dr. Koichiro Otani for his initial input in adapting the Consumer Assessment of Healthcare Providers and Systems (CAHPS) survey instrument and his contribution to the discussion on the project design. We acknowledge the clinical contribution of Dr. Sarah Zalwango and the help of the research assistants; Damalie Nakkonde, Esther Nakayenga and Joyce Nalubowa in collecting the data, the staff at the HIV/AIDS clinic, the data clerk and the participants in Kampala, Uganda.

## Author Contributions

**Conceptualization:** Juliet Nabbuye Sekandi, Robert Kakaire, Simon Mutembo, Jane Namangolwa Mutanga.

**Data curation:** Maria Eugenia Castellanos, Henok Woldu, Robert Kakaire.

**Formal analysis:** Juliet Nabbuye Sekandi, Maria Eugenia Castellanos, Henok Woldu, Jane Namangolwa Mutanga.

**Methodology:** Juliet Nabbuye Sekandi, Henok Woldu, Simon Mutembo, Jane Namangolwa Mutanga.

**Project administration:** Juliet Nabbuye Sekandi, Robert Kakaire.

**Supervision:** Juliet Nabbuye Sekandi.

**Validation:** Juliet Nabbuye Sekandi, Maria Eugenia Castellanos, Henok Woldu, Jane Namangolwa Mutanga.

**Visualization:** Maria Eugenia Castellanos, Henok Woldu, Simon Mutembo, Jane Namangolwa Mutanga.

**Writing – original draft:** Juliet Nabbuye Sekandi, Maria Eugenia Castellanos, Henok Woldu, Robert Kakaire, Simon Mutembo, Jane Namangolwa Mutanga.

**Writing – review & editing:** Juliet Nabbuye Sekandi, Maria Eugenia Castellanos, Henok Woldu, Robert Kakaire, Simon Mutembo, Jane Namangolwa Mutanga.

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
