## [Decision Letter · Decision Letter 0]

26 May 2022

PONE-D-21-27071Health Provider Skills and Quality of Reception Services Influence Patient Satisfaction Among People Living with HIV Receiving Antiretroviral Therapy in Urban Uganda: A Factor AnalysisPLOS ONE

Dear Dr. Sekandi,

Thank you for submitting your manuscript to PLOS ONE. After careful consideration, we feel that it has merit but does not fully meet PLOS ONE’s publication criteria as it currently stands. Therefore, we invite you to submit a revised version of the manuscript that addresses the points raised during the review process.

Please see the reviewer's assessment copied below. When revising your manuscript, please in particular ensure you fully address the reviewer's comments with regards to the modification and scoring of the CAHPS survey. Additionally, please provide with your revised manuscript, as supplementary files, the questionnaires used in this study, both in Luganda and in English.

We look forward to receiving your revised manuscript.

Kind regards,

Hugh Cowley

Senior Editor

PLOS ONE

Journal Requirements:

3. Thank you for stating the following financial disclosure: 'JNS and KO received an internal seed grant from the University of Georgia.

We note that one or more of the authors is affiliated with the funding organization, indicating the funder may have had some role in the design, data collection, analysis or preparation of your manuscript for publication; in other words, the funder played an indirect role through the participation of the co-authors. If the funding organization did not play a role in the study design, data collection and analysis, decision to publish, or preparation of the manuscript and only provided financial support in the form of authors' salaries and/or research materials, please do the following:

a. Review your statements relating to the author contributions, and ensure you have specifically and accurately indicated the role(s) that these authors had in your study. These amendments should be made in the online form.

b. Confirm in your cover letter that you agree with the following statement, and we will change the online submission form on your behalf: 

“The funder provided support in the form of salaries for authors [insert relevant initials], but did not have any additional role in the study design, data collection and analysis, decision to publish, or preparation of the manuscript. The specific roles of these authors are articulated in the ‘author contributions’ section.

"Funding: This work was supported by the University of Georgia internal seed grants program. 

The authors are grateful to Dr. Koichiro Otani for his initial assistance in adopting the Consumer Assessment of Healthcare Providers and Systems (CAHPS) survey instrument for measuring satisfaction. We acknowledge the clinical contribution of Dr. Sarah Zalwango and the help of the research assistants; Damalie Nakkonde, Esther Nakayenga and Joyce Nalubowa in collecting the data, the staff at the HIV clinic, the data clerk and the participants in Kampala, Uganda.

AUTHORSHIP CONTRIBUTIONS AND CONFIRMATION STATEMENT 

Conception and design: JNS, JM, SM, RK data collection: RK, JS data analyses: MEC, HW, JNS interpretation of the study results: JNS, MEC, HW, RK, JM, SM first draft of the manuscript: JNS, MEC. Review and edit of the manuscript: JNS, MEC, HW, RK, SM, JM. All authors approved the final version of the manuscript."

Please remove any funding-related text from the manuscript and let us know how you would like to update your Funding Statement. Currently, your Funding Statement reads as follows: "JNS and KO received an internal seed grant from the University of Georgia.

Reviewers' comments:

Reviewer's Responses to Questions

**Comments to the Author**

1. Is the manuscript technically sound, and do the data support the conclusions?

Reviewer #1: Yes

2. Has the statistical analysis been performed appropriately and rigorously? 

Reviewer #1: Yes

3. Have the authors made all data underlying the findings in their manuscript fully available?

Reviewer #1: Yes

4. Is the manuscript presented in an intelligible fashion and written in standard English?

Reviewer #1: Yes

5. Review Comments to the Author

Reviewer #1: Thank you for the opportunity to review manuscript PONE-D-21-27071 entitled, S Health Provider Skills and Quality of Reception Services Influence Patient Satisfaction Among People Living with HIV Receiving Antiretroviral Therapy in Urban Uganda: A Factor Analysis

The study reports findings from a cross sectional study conducted among people with HIV to evaluate the internal reliability of the CAHPS scale and factors associated with higher patient satisfaction. The paper is generally well-written and well-organized. The description of procedures for translating the instruments and evaluating its internal validity and reliability are generally well-conceived and carried out. The study’s premise could be argued better. It’s not clear why the new instrument is needed. Likewise, the rationale and procedures for adapting the instrument are less clear and instructions for scoring are not clear, which might discourage future use of the scale. The finding that facility satisfaction was higher in those who reported higher satisfaction with receptionists is interesting.

Authors state the “CAHPS has not been assessed in an African population”. While this may be true, it’s not clear why this justifies the study. In arguing the study’s premise and rationale for the current study, it may be beneficial to discuss the research studies that have evaluated patient satisfaction in SSA and the strengths and limitations of those instruments. For example, researchers in Lusaka recently used a 9-item satisfaction scale adapted from the Adult Primary Care Questionnaire: Mukamba N, Chilyabanyama ON, Beres LK…Schwartz SR. Patients’ satisfaction with HIV care providers in public health facilities in Lusaka: a study of patients who were lost-to-follow-up from HIV care and treatment. AIDS and Behavior. 2020 Apr;24(4):1151-60.

Page 4: The authors discuss their sampling approach and describe using “a sampling interval of five”. One might assume that every fifth patient was invited to participate, however, more details about the process and personnel (clinic staff or researchers) who made the initial contact to recruit subjects and determine eligibility. Also, how was social desirability bias addressed given that participants were being asked to rate the same facility where they received services?

Page 5: The authors describe modifying/adapting the CAHPS instrument and selecting 18 questions. What was the rationale for modifying the instrument? It might be useful to mention the number of items in the original instrument and the rational for selecting 18. Also, rather than list the items individually in a paragraph, it might be preferable to have a table. Was mental and emotional health part of the original survey? How were these constructs measured and why were they important to include. What other questions were added or dropped before administering the modified instrument?

Page 5-6: The formula for scoring the CAHPS is unclear. It’s also unclear whether the authors are using the original scoring formula or their own.

The significance of the patient satisfaction scores are hard to appreciate without an understanding of the scale range and what different values represent. The authors stated previously that the maximum possible score for the “composite index” was 3.94, which is not easily interpreted. For example, it’s unclear what a “2.08” means for “How often did you see a provider within one hour”.

It would be interesting to see the actual survey responses. For example, the percentage who agreed or disagreed with each of the 18 items.

The word “prediction” is used frequently in describing the multivariate modeling and results. Given the study design and the lack of a conceptual model, might be preferable to talk about “associations” in the data. The authors do a good job of acknowledging this in the discussion, but not as much with the results.

From the output in Table 4, it appears that sex and mental health were associated with health facility rating as were patient satisfaction sub-scales. It’s not clear where the authors see evidence of an association with educational level, although this is mentioned in text.

Line 280-281: The authors are cautioned to avoid language implying that factor loadings are indicative of the influence of these items of patient satisfaction. The loadings indicate how well the items cluster.

Minor

Abstract

Consider using a comma after the phrase “emotional health”.

Manuscript

line 65 – the “S” in AIDS should be capitalized.

line 66, 67 – capitalization not needed for “Antiretroviral”, “People”, “Living”.

line 71 = comma after “Africa”

line 113, add the word “as” after “such”.

line 181, remove the word “and”.

Alpha reported variously as 0.92 and 0.94. Check to make sure reliability estimates are reported consistently

6. PLOS authors have the option to publish the peer review history of their article (what does this mean?). If published, this will include your full peer review and any attached files.

Reviewer #1: No

---

## [Author Response · Author response to Decision Letter 0]

9 Aug 2022

Reviewer #1: 

Thank you for the opportunity to review manuscript PONE-D-21-27071 entitled, S Health Provider Skills and Quality of Reception Services Influence Patient Satisfaction Among People Living with HIV Receiving Antiretroviral Therapy in Urban Uganda: A Factor Analysis

The study reports findings from a cross sectional study conducted among people with HIV to evaluate the internal reliability of the CAHPS scale and factors associated with higher patient satisfaction. The paper is generally well-written and well-organized. The description of procedures for translating the instruments and evaluating its internal validity and reliability are generally well-conceived and carried out. The study’s premise could be argued better. It’s not clear why the new instrument is needed. Likewise, the rationale and procedures for adapting the instrument are less clear and instructions for scoring are not clear, which might discourage future use of the scale. The finding that facility satisfaction was higher in those who reported higher satisfaction with receptionists is interesting.

Response: The CG-CAHPS instrument was use because of its focus on capturing patient experiences. The CAHPS tool is unique in that it covers domains of patient experience: accessibility to care, communication with providers, care coordination and interaction with the non-medical staff i.e. receptionists/clerks. 

Authors state the “CAHPS has not been assessed in an African population”. While this may be true, it’s not clear why this justifies the study. In arguing the study’s premise and rationale for the current study, it may be beneficial to discuss the research studies that have evaluated patient satisfaction in SSA and the strengths and limitations of those instruments. For example, researchers in Lusaka recently used a 9-item satisfaction scale adapted from the Adult Primary Care Questionnaire: Mukamba N, Chilyabanyama ON, Beres LK…Schwartz SR. Patients’ satisfaction with HIV care providers in public health facilities in Lusaka: a study of patients who were lost-to-follow-up from HIV care and treatment. AIDS and Behavior. 2020 Apr;24(4):1151-60.

Response: There is no standard to measure patient satisfaction, however a variety of scales exist and have been used in HIV-infected populations in the African setting. For example, Mukamba and colleagues (2020) adapted the Adult Primary Care Questionnaire with 9-items that focus specifically on the patients’ interaction with the doctor. In our study, we chose to use the CAHPS instrument because it captures experiences with a broader range health staff including the clerks and receptionists who might indeed influence their satisfaction with the services. Understanding patients experiences from check-in to checkout of the health facility is essential to providing a complete picture of health care quality. Moreover, capturing the non-medical aspects of a patients experience at a clinic visit would ensure that the health quality interventions encompass the full spectrum of touch points. 

Page 4: The authors discuss their sampling approach and describe using “a sampling interval of five”. One might assume that every fifth patient was invited to participate, however, more details about the process and personnel (clinic staff or researchers) who made the initial contact to recruit subjects and determine eligibility. Also, how was social desirability bias addressed given that participants were being asked to rate the same facility where they received services?

Response: The number of patients to be interviewed were based on the daily average patient load in the study clinic. Based on the records, approximately 113 patients visited the HIV clinic each day. We aimed to recruit 20 patients per day in order to achieve the estimated sample size within the planned study enrollment period. We then employed systematic random sampling of every 5th patients who showed up at the clinic on each day of week, Monday through Friday. The clinic nurses assisted the study to team to identify eligible patients and referred them for brief information about the study and then invited to participate voluntarily. If they declined they were replaced by the next patient who met the interval of 5 sampling procedure. 

Response: Although social desirability bias could have possibly occurred given that participants were being asked to rate the same facility where they received services, we minimized it by hiring a separate research team of nurses who do not routinely work at the same HIV clinic. Additionally, patients were assured that their information was to be kept confidential and only shared in a general report as recommendations for improvement of quality of services.

Page 5: The authors describe modifying/adapting the CAHPS instrument and selecting 18 questions. What was the rationale for modifying the instrument? It might be useful to mention the number of items in the original instrument and the rational for selecting 18. Also, rather than list the items individually in a paragraph, it might be preferable to have a table. Was mental and emotional health part of the original survey? How were these constructs measured and why were they important to include? What other questions were added or dropped before administering the modified instrument?

Response: Questions #1-5 were modified to suit the context and questions #6- 24 were adapted for use in specifically measuring satisfaction with the domains of interest. A table of all the items adapted from the CAHPS-CG (Table 1) is now shown in the text and a copy of the version of the modified instrument is provided in supporting information as S3.

Page 5-6: The formula for scoring the CAHPS is unclear. It’s also unclear whether the authors are using the original scoring formula or their own. 

Response: We have used specific guidance from the CAHPS to normalize the scoring of the CAHPS. See link below:

The CAHPS Ambulatory Care Improvement Guide: Determining Where to Focus Efforts to Improve Patient Experience (ahrq.gov) see Figure 5-2. Comparison of Practice Site Normalized Mean Scores to Group Normalized Mean Scores

Normalizing is a way to transform all scores to the same scale, typically 0 to 100. It is done to ease comparison across items and composites that use different response scales. To transform the scores, one would first transform the response values at the respondent level from 0-100 using the following formula: Normalized Score = 100*(Respondent’s selected response value – Minimum response value on scale) / (Maximum response value – Minimum response value) For example, the responses on a four-point scale would be normalized as follows: Response Option 1, 2, 3, 4 and Normalized Response 0.00,33.33, 66.67, 100.00 respectively.

The significance of the patient satisfaction scores are hard to appreciate without an understanding of the scale range and what different values represent. The authors stated previously that the maximum possible score for the “composite index” was 3.94, which is not easily interpreted. For example, it’s unclear what a “2.08” means for “How often did you see a provider within one hour”. 

Response: The patient satisfaction scores have been normalized to a scale of 0 to 100 and an arithmetic mean calculated to make it easier for the readers to understand.

It would be interesting to see the actual survey responses. For example, the percentage who agreed or disagreed with each of the 18 items. 

Response: The results of responses from the 18 items have been presented in figure 1 and in a table as supporting information see Table S1.

The word “prediction” is used frequently in describing the multivariate modeling and results. Given the study design and the lack of a conceptual model, might be preferable to talk about “associations” in the data. The authors do a good job of acknowledging this in the discussion, but not as much with the results. 

Response: This language has been corrected to show that they are associations

From the output in Table 4, it appears that sex and mental health were associated with health facility rating as were patient satisfaction sub-scales. It’s not clear where the authors see evidence of an association with educational level, although this is mentioned in text.

Response: This has been clarified to show that education at senior level was a significant factor at bivariate analysis but was not significant at multivariable regression analysis suggesting it is an independent factor.

Line 280-281: The authors are cautioned to avoid language implying that factor loadings are indicative of the influence of these items of patient satisfaction. The loadings indicate how well the items cluster. 

Response: This language has been modified to read as such “Factor loadings specifically show how well the items cluster together. Our findings highlight the importance of the timeliness with reception services and courteous treatment of patients by the clerical staff in the HIV clinics. ” now in line 301- 303.

Minor

Abstract

Consider using a comma after the phrase “emotional health”.- This has been corrected

Manuscript

line 65 – the “S” in AIDS should be capitalized. - This has been corrected

Manuscript

line 66, 67 – capitalization not needed for “Antiretroviral”, “People”, “Living”. This has been corrected

Manuscript

line 71 = comma after “Africa” This has been corrected

Manuscript

line 113, add the word “as” after “such”. This has been corrected

Manuscript

line 181, remove the word “and”. This has been corrected

Manuscript

Alpha reported variously as 0.92 and 0.94. Check to make sure reliability estimates are reported consistently- This has been corrected, 0.94 is the overall Cronbach’s alpha number

---

## [Decision Letter · Decision Letter 1]

10 Oct 2022

PONE-D-21-27071R1Health Provider Skills and Quality of Reception Services Influence Patient Satisfaction Among Persons Living with HIV and Receiving Antiretroviral Therapy in Urban Uganda: A Factor AnalysisPLOS ONE

Dear Dr. Juliet N Sekandi

Thank you for submitting your manuscript to PLOS ONE. After careful consideration, we feel that it has merit but does not fully meet PLOS ONE’s publication criteria as it currently stands. Therefore, we invite you to submit a revised version of the manuscript that addresses the points raised during the review process.

We look forward to receiving your revised manuscript.

Kind regards,

Armin Aryannejad, M.D.

Guest Editor

PLOS ONE

Reviewers' comments:

Reviewer's Responses to Questions

**Comments to the Author**

1. If the authors have adequately addressed your comments raised in a previous round of review and you feel that this manuscript is now acceptable for publication, you may indicate that here to bypass the “Comments to the Author” section, enter your conflict of interest statement in the “Confidential to Editor” section, and submit your "Accept" recommendation.

Reviewer #1: All comments have been addressed

Reviewer #2: All comments have been addressed

Reviewer #3: (No Response)

2. Is the manuscript technically sound, and do the data support the conclusions?

Reviewer #1: (No Response)

Reviewer #2: Yes

Reviewer #3: Yes

3. Has the statistical analysis been performed appropriately and rigorously? 

Reviewer #1: (No Response)

Reviewer #2: Yes

Reviewer #3: Yes

4. Have the authors made all data underlying the findings in their manuscript fully available?

Reviewer #1: (No Response)

Reviewer #2: Yes

Reviewer #3: No

5. Is the manuscript presented in an intelligible fashion and written in standard English?

Reviewer #1: (No Response)

Reviewer #2: Yes

Reviewer #3: Yes

6. Review Comments to the Author

Reviewer #1: (No Response)

Reviewer #2: Thanks for sending me this interesting survey on satisfactin of HIV patients with outpatient clinics. Here are some comments.

This is a revised manuscript and as seen in track-change file, substantial modifications have been carried out. I believe this survey is important especially in developing countries with volatile healthcare atmosphere.

The questionnaire name is inconsistently used throughout paper. CAHPS vs CHAPS

The introduction is prepared very well

The methods sections is very long and many details could be shortened or merged.

The note that about ¾ of participants were female could jeopardize the sampling method. Authors need to mention the baseline composition of whole clinic patients and see if the participants are similarly ¾ of whole pool are females, too.

Beside the recall bias, sampling bias / selection bias should be mentioned in limitation section too. For instance, patients being less interested in this clinic or those with worse conditions are less likely to participate.

Reviewer #3: I would like to thank the editor for providing the chance to review this submission and the authors for they precious efforts. The authors of this study investigated the important notion of patient satisfaction among patients with HIV residing in urban Uganda as a health services research endeavor. The submission benefits from a previous round of review and revision and I am happy to evaluate the R1 draft. In my initial evaluation, the authors were successful in addressing and answering the previous reviewer’s comments in this revision. However, the manuscript could benefit from further changes. My comments and suggestions in this regard on this version of the manuscript are provided below.

1. Lines 1-2: the provided title could be briefer and clearer in transferring the study notion.

2. Lines 31-33: this part of the introduction is more a methods part and could be moved to the next part in the abstract and instead the aim of study is suggested to be added in these lines.

3. Line 35: since none of the authors of this study are affiliated with any institution in Uganda, it should be clear that authors used a pre-collected data or they conducted themselves the data collection for this study.

4. Lines 43-47: adding some numbers to the provided results in abstract could be beneficial for readers, especially about the significant associations of mentioned factors with patient satisfaction.

5. Lines 50-52: before ant general conclusion, providing a sentence driven by the results of this study is necessary to summarize the findings of this investigation.

6. Introduction: revising this section in order to make the first paragraph talking about the epidemiology and burden of HIV/AIDS in general and in Uganda, and the second paragraph talking about patient satisfaction analysis could be better and is highly suggested.

7. Line 122: it should be made clear all through the text that this study recruited all HIV/AIDS patients and not only those in HIV stage. This is vague in the current format of presentation.

8. Lines 219-221: the authors may add about the logic of choosing independent variables for the regression analysis. Were the mentioned variables included based on a literature review or by significant results of a primary bivariate regression analysis? The details of these steps should be added to this part of the methods.

9. Line 238: adding the details of statistical measures like mean and SD used to report the results to the methods could be useful.

10. Discussion: this section needs a paragraph on the practical implications of this study findings and some points for local and regional health authorities to benefit the most by the conducted efforts.

7. PLOS authors have the option to publish the peer review history of their article (what does this mean?). If published, this will include your full peer review and any attached files.

Reviewer #1: **Yes: **Gabriel Culbert

Reviewer #2: **Yes: **Esmaeil Mohammadi, MD MPH

Reviewer #3: **Yes: **Sina Azadnajafabad, MD, MPH

---

## [Author Response · Author response to Decision Letter 1]

22 Nov 2022

Response to Reviewers Comments

Reviewer #2: Thanks for sending me this interesting survey on satisfaction of HIV patients with outpatient clinics. Here are some comments.

This is a revised manuscript and as seen in track-change file, substantial modifications have been carried out. I believe this survey is important especially in developing countries with volatile healthcare atmosphere.

1. The questionnaire name is inconsistently used throughout paper. CAHPS vs CHAPS—

Response: This error has been corrected to be consistent as CAHPS

The introduction is prepared very well

2.The methods sections is very long and many details could be shortened or merged.

Response: This point is well noted and appreciated. However, the authors have agreed to retain the details in the methods section because some content was just added to expand details as part of the first revision in response to comments from other reviewers.

3. The note that about ¾ of participants were female could jeopardize the sampling method. Authors need to mention the baseline composition of whole clinic patients and see if the participants are similarly ¾ of whole pool are females, too.

Response: We acknowledge the possible overrepresentation of female participants in the study sample. However, we note that the demographic distribution of HIV prevalence is Uganda is consistently skewed towards female. In 2018, the Uganda HIV/AIDS progress report showed that the prevalence of HIV/AIDS in women 15+ is 7.6% while that of men is 4.7% in the same age group. Therefore, based on this distribution the proportion of female patients attending HIV/AIDS clinic can be expected to be higher accordingly. 

4. Beside the recall bias, sampling bias / selection bias should be mentioned in limitation section too. For instance, patients being less interested in this clinic or those with worse conditions are less likely to participate.

Response: We have added sampling and selection bias as part of the acknowledged possible study limitations 

Reviewer #3: I would like to thank the editor for providing the chance to review this submission and the authors for their precious efforts. The authors of this study investigated the important notion of patient satisfaction among patients with HIV residing in urban Uganda as a health services research endeavor. The submission benefits from a previous round of review and revision and I am happy to evaluate the R1 draft. In my initial evaluation, the authors were successful in addressing and answering the previous reviewer’s comments in this revision. However, the manuscript could benefit from further changes. My comments and suggestions in this regard on this version of the manuscript are provided below.

1. Lines 1-2: the provided title could be briefer and clearer in transferring the study notion.

Response: The title has been shortened as suggested by the reviewer to read “Patient satisfaction among persons living with HIV and receiving antiretroviral therapy in urban Uganda: A factor analysis”

2. Lines 31-33: this part of the introduction is more a methods part and could be moved to the next part in the abstract and instead the aim of study is suggested to be added in these lines.

Response: This suggested revision has been made. Thank you!

3. Line 35: since none of the authors of this study are affiliated with any institution in Uganda, it should be clear that authors used a pre-collected data or they conducted themselves the data collection for this study. 

Response: The first (Juliet N. Sekandi) and fourth (Robert Kakaire) authors, are both native Ugandans who both physically conducted this project during their post-doc and doctoral training respectively in summer time in Uganda. They led the development and data collection while supervising the research assistants. None of the research assistants fully met the requirements to be co-author hence they are listed in the acknowledgements.

4. Lines 43-47: adding some numbers to the provided results in abstract could be beneficial for readers, especially about the significant associations of mentioned factors with patient satisfaction. 

Response: The p-values to indicate significant associations have been added to the abstract

5. Lines 50-52: before ant general conclusion, providing a sentence driven by the results of this study is necessary to summarize the findings of this investigation. 

Response: The conclusion has been revised according to the reviewer’s suggestion, Thank you!

6. Introduction: revising this section in order to make the first paragraph talking about the epidemiology and burden of HIV/AIDS in general and in Uganda, and the second paragraph talking about patient satisfaction analysis could be better and is highly suggested. 

Response: We appreciate this suggestion and we have made some changes. However, we note that the current version of the introduction was set up in that format based on a previous revision suggested by another reviewer. To avoid any confusion the authors have tried to strike a balance in the revisions.

7. Line 122: it should be made clear all through the text that this study recruited all HIV/AIDS patients and not only those in HIV stage. This is vague in the current format of presentation. 

Response: This presentation has been corrected to HIV/AIDS consistently throughout the manuscript.

8. Lines 219-221: the authors may add about the logic of choosing independent variables for the regression analysis. Were the mentioned variables included based on a literature review or by significant results of a primary bivariate regression analysis? The details of these steps should be added to this part of the methods. 

Response: Yes, we based primarily on the significant variables at bivariate analysis and as the literature review including our study in Kampala as the second guide to selection. This has been added to the manuscript

9. Line 238: adding the details of statistical measures like mean and SD used to report the results to the methods could be useful. 

Response: These are already included

10. Discussion: this section needs a paragraph on the practical implications of this study findings and some points for local and regional health authorities to benefit the most by the conducted efforts. 

Response: A short paragraph on practical implications has been added to the discussion section

---

## [Decision Letter · Decision Letter 2]

8 Jan 2023

Patient Satisfaction among Persons Living with HIV/AIDS and Receiving Antiretroviral Therapy in Urban Uganda: A factor Analysis

PONE-D-21-27071R2

Dear Dr. Juliet N Sekandi

We’re pleased to inform you that your manuscript has been judged scientifically suitable for publication and will be formally accepted for publication once it meets all outstanding technical requirements.

Kind regards,

Armin Aryannejad, M.D.

Guest Editor

PLOS ONE

Additional Editor Comments (optional):

Reviewers' comments:

Reviewer's Responses to Questions

**Comments to the Author**

1. If the authors have adequately addressed your comments raised in a previous round of review and you feel that this manuscript is now acceptable for publication, you may indicate that here to bypass the “Comments to the Author” section, enter your conflict of interest statement in the “Confidential to Editor” section, and submit your "Accept" recommendation.

Reviewer #2: All comments have been addressed

Reviewer #3: All comments have been addressed

2. Is the manuscript technically sound, and do the data support the conclusions?

Reviewer #2: Yes

Reviewer #3: Yes

3. Has the statistical analysis been performed appropriately and rigorously? 

Reviewer #2: Yes

Reviewer #3: Yes

4. Have the authors made all data underlying the findings in their manuscript fully available?

Reviewer #2: Yes

Reviewer #3: No

5. Is the manuscript presented in an intelligible fashion and written in standard English?

Reviewer #2: Yes

Reviewer #3: Yes

6. Review Comments to the Author

Reviewer #2: I believe all my comments are being addressed and I have no further concerns. congratulations to the authors for their achievement!

Reviewer #3: Thanks for the revised draft. All comments were addressed adequately and soundly. The current format of the manuscript meets the publication criteria at PLOS One in my opinion.

7. PLOS authors have the option to publish the peer review history of their article (what does this mean?). If published, this will include your full peer review and any attached files.

Reviewer #2: **Yes: **Esmaeil Mohammadi, MD MPH

Reviewer #3: **Yes: **Sina Azadnajafabad, MD, MPH

---

## [Editor Report · Acceptance letter]

13 Jan 2023

PONE-D-21-27071R2 

Patient satisfaction among persons living with HIV/AIDS and receiving antiretroviral therapy in urban Uganda: A factor analysis 

Dear Dr. Sekandi:

I'm pleased to inform you that your manuscript has been deemed suitable for publication in PLOS ONE. Congratulations! Your manuscript is now with our production department. 

Kind regards, 

on behalf of

Dr. Armin Aryannejad 

Guest Editor

PLOS ONE